# The Pattern of Blood–Milk Exchange for Antiparasitic Drugs in Dairy Ruminants

**DOI:** 10.3390/ani11102758

**Published:** 2021-09-22

**Authors:** Fernanda Imperiale, Carlos Lanusse

**Affiliations:** Laboratorio de Farmacología, Centro de Investigación Veterinaria de Tandil (CIVETAN, CONICET-CICPBA), Facultad de Cs. Veterinarias, UNCPBA, Tandil 7000, Argentina; clanusse@vet.unicen.edu.ar

**Keywords:** dairy animals, antiparasitic drugs, rational use in parasite control, plasma–milk exchange, residues in milk dairy products

## Abstract

**Simple Summary:**

This review article is focused on the description of the plasma–milk partition coefficients for different antiparasitic drug classes in dairy ruminants, and it contributes to rational pharmaco-therapy in lactating dairy animals, which is critical to understand the pattern of drug excretion in milk as well as the residual concentration patterns in dairy products elaborated by processing milk from drug-treated animals.

**Abstract:**

The prolonged persistence of milk residual concentration of different antiparasitic drugs in lactating dairy animals should be considered before recommending their use (label or extra-label) for parasite control in dairy animals. The partition blood-to-milk ratio for different antiparasitic compounds depends on their ability to diffuse across the mammary gland epithelium. The high lipophilicity of some of the most widely used antiparasitic drugs explains their high partition into milk and the extended persistence of high residual concentrations in milk after treatment. Most of the antiparasitic drug compounds studied were shown to be stable in various milk-related industrial processes. Thus, the levels of residues detected in raw milk can be directly applicable to estimating consumer exposure and dietary intake calculations when consuming heat-processed fluid milk. However, after milk is processed to obtain milk products such as cheese, yogurt, ricotta, and butter, the residues of lipophilic antiparasitic drugs are higher than those measured in the milk used for their elaboration. This review article contributes pharmacokinetics-based information, which is useful to understand the relevance of rational drug-based parasite control in lactating dairy ruminants to avoid undesirable consequences of residual drug concentrations in milk and derived products intended for human consumption.

## 1. Introduction

Parasitism infections are frequently subclinical in adult animals and are often connected with lower levels of milk yield [1,2,3,4,5]. After strategic anthelmintic treatment, naturally infected lactating dairy cows, sheep, and goats have shown an enhancement in milk production [6,7,8,9,10]. Several management strategies are used to prevent or minimize production losses; however, the use of antiparasitic drugs is still the main tool available against parasitic diseases in lactating dairy animals.

Absorption, distribution, metabolism and excretion are the physiological processes that govern the time path of drug fate in the body. Antiparasitic drugs are currently administered to dairy animals in the form of injectable, oral drench, and topical preparations. As shown in Figure 1, after administration of an antiparasitic formulation, the drug must be released from the vehicle where it is formulated, and once soluble at the site of administration, it is absorbed into systemic circulation and must reach its site of action. The efficacy of antiparasitic drugs is related to their pharmacokinetic and pharmacodynamic behavior in the body of the treated animal. The distribution process from bloodstream (pH = 7.4) to peripheral tissues—that is, the mammary gland (milk pH = 6.5)—is dependent on the physicochemical properties of the compound (pKa, lipid solubility, molecular weight), the concentration gradient between blood and tissue, the pH of the medium between the sides of the membrane, and the chemical’s affinity for tissue constituents. The physicochemical properties of the chemical are most important in determining its affinity for distribution to a specific tissue. For most molecules, distribution out of the blood into tissues, including the mammary gland, is by simple diffusion down a concentration gradient.

The mammary gland epithelium, like other biologic membranes, acts as a lipid barrier, and the high lipophilicity of the compounds favors partitioning into milk (see Figure 1). Therefore, the rate of diffusion of the compound across a membrane is directly proportional to its concentration gradient across the membrane, lipid/water partition coefficient, and diffusion coefficient [11]. Hydrophilic drug compounds have only limited access to the mammary gland. However, high lipophilicity accounts for extensive plasma-to-milk exchange, which accounts for the high excretion of drugs in milk (see Figure 1). This has been clearly shown for some lipophilic antiparasitic drugs such as moxidectin (MXD) [12], where long persistence of residues in milk and the greater amount of MXD excreted in milk has been shown in comparison with similar but less lipid-soluble compounds.

In addition to simple diffusion down a concentration gradient, the cellular efflux of many drugs through ATP-binding cassette (ABC) transporters, among others, protects against the toxicity of xenobiotics and influences pharmacokinetics. The breast cancer resistance protein (BCRP/ABCG2) is also highly expressed during lactation in the mammary gland tissues, thereby playing a central role in the active secretion of various xenobiotics into milk [13,14]. The BCRP transporter has obvious implications about the presence of residues in milk due to the active secretion of drugs and xenotoxins into milk. Many reports have shown that drug–drug interactions that inhibit mammary BCRP can influence drug secretion into milk and consequently the accumulation of drug residues in milk [15,16,17].

After anthelmintic treatment of lactating dairy animals, the risks associated with residues in milk meant for human consumption and milk products may be present and should be considered [18,19,20,21]. However, in endemic areas of the world where the parasitic diseases such as fasciolosis, haemonchosis, and ectoparasites are recognized as a major problem in dairy animals, parasite control programs are often implemented as extra-label antihelmintic treatments [22,23].

Currently, the maximum allowable concentrations of medicinal residues in foodstuffs of animal origin, named maximum residue limits (MRLs), have been set by federal authorities in various countries for some antiparasitics, such as albendazole, oxfendazole, fenbendazole, thiabendazole, triclabendazole, closantel, nitroxynil, oxyclozanide, eprinomectin, moxidectin, and ivermectin, in milk for human consumption or manufacturing purposes. Therefore, for all licensed antiparasitic drugs administered to livestock that produce food for human consumption, setting MRLs and withdrawal periods is a necessary step, and these withdrawal periods must be strictly observed. Withdrawal time is the time period (hours, days, week, etc.) that should elapse after the last administration of a drug in order to avoid residual concentration above the stipulated MRL or achieve zero concentration for drugs with no MRL established [24,25].

The current review article discusses the pharmacokinetic properties of various antiparasitic drug classes that are widely used for parasite control in dairy animals, including their plasma-to-milk partition coefficients, and describes the distribution of drug residues between milk and milk products produced by processing milk from treated animals.

## 2. Pattern of Antiparasitic Drugs Milk Excretion in Dairy Ruminants

### 2.1. Benzimidazoles

Benzimidazole (BZD) anthelmintics are widely used in veterinary medicine. They are currently marketed as broad-spectrum anthelmintics for the control of gastrointestinal nematodes, lungworms, tapeworms, and liver fluke. The most important restriction for the formulation of BZD methylcarbamate anthelmintics, such as albendazole (ABZ), fenbendazole (FBZ), etc., is the low solubility in aqueous medium. Therefore, BZD is formulated as an oral and intraruminal suspension [26]. The development of injectable formulations such as aqueous solution of albendazole sulphoxide (ABZSO) and oxfendazole (OFZ) has overcome the low systemic availability of these compounds. Release from the dosage form and absorption precede the entry of BZD into the bloodstream, which serves as the vehicle by which drug/metabolite molecules are distributed to different tissues of the body, including the mammary gland. The pharmacokinetics, metabolism, and tissue distribution of BZD compounds have been widely described in ruminants [27,28,29,30,31,32]. Usually, in ruminant species, BZD compounds show a relatively high volume of distribution, protein binding of less than 50% in plasma, and a relatively fast elimination rate. Moreover, the pattern of milk excretion for some orally administered BZD compounds in lactating dairy cattle [33,34,35], goats [36], and sheep [37] and parenterally administered BZD anthelmintic in lactating dairy cows [35]has been reported.

Many reports have shown that after oral administration of ABZ to lactating dairy cows (5 mg/kg), sheep (12.5 mg/kg), and goats (7.5 mg/kg), residues of ABZSO and albendazole sulphone (ABZSO2) metabolites were found in milk, while the parent drug (ABZ) was not detected. Sulpho-metabolites were measured at the first and second milking. The ABZSO2 (0.86 µg/mL) concentration was higher than ABZSO (0.28 µg/mL) concentration in cow’s milk 12 h post-treatment. Both residual concentrations decreased markedly 24 h post-treatment (second milking) and were lower than ABZSO residues. Despite its rapid elimination, ABZO2 was detected at very low concentrations in milk at the 13th milking (156 h) post-treatment. Similarly, Fletouris et al. [38] reported that ABZSO2 showed the highest milk residue level but it was measured for a longer period, possibly due to the type of administration (bolus) and the higher dose administered. Moreover, the 2-aminosulfone metabolite, the N-deacetylation product of the sulphone metabolite, appeared at low concentrations 12 h after treatment, reaching its maximum value more slowly (at 24 h or 36 h), and also disappeared more slowly (at about 192 h). Similarly, after oral administration of ABZ to goats (7.5 mg/kg) and sheep (12.5 mg/kg), ABZSO and ABZSO2 metabolites were found at high levels in milk collected within 24 h after treatment [36,37]. From the third day, the ABZ metabolites in milk were lower than the MRL (as the sum of all metabolites) established for ABZ (100 µg/kg) [39].

After subcutaneous (sc) administration of ABZSO to lactating dairy cows (3 mg/kg), sulpho-metabolite (ABZSO and ABZSO2) residues were also found in milk [35]. Both ABZSO and ABZSO2 residues were lower than those obtained after ABZ oral treatment [35].Moreover, ABZSO was the highest residue of the sulpho-metabolite retrieved in milk (0.18 µg/mL) 12 h post-treatment. After that, the milk concentrations markedly decreased 24 h post-treatment; however, ABZSO continued to be the highest milk residue.

After oral administration of OFZ (5 mg/kg) to dairy cows, milk residual concentration profiles of OFZ, fenbendazole sulphone (FBZSO2), and FBZ were reported [35]. The highest concentrations were obtained for OFZ (0.39 µg/mL) 12 h post-treatment. Furthermore, FBZSO2 reached its maximum concentration (0.17 µg/mL) later (36 h), and both analytes were measured in milk up to 72 h post-treatment. FBZ, the parent drug, reached the lowest concentrations in milk. FBZ is a thioether compound produced by sulpho-reduction of OFZ in the rumen and intestine [40,41,42], and it attained the highest concentration level 24 h post-treatment (0.10 µg/mL), which can be detected in milk up to 48 h post-treatment.

After sc administration of OFZ (3 mg/kg) to dairy cows as described by Moreno et al. [35], OFZ and FBZSO2 milk residual concentrations were obtained. In contrast to the oral route, the parent drug (FBZ) was not found in milk. FBZSO2 was the main analyte retrieved from milk, reaching the maximum milk residue level (0.042 µg/mL) 36 h after treatment. The OFZ concentration was the highest (0.03 µg/mL) at the first milking, and it was detected in milk up to 48–60 h post-treatment. From day 5, the OFZ metabolites in milk were lower than the MRL (as the sum of extractable residues can be oxidized to FBZSO2) established for OFZ (10 µg/kg) [43].

After administering FBZ as an oral suspension (5 mg/kg) to dairy cows, anthelmintic drug residues in milk were reported, reaching the maximum concentration in milk 24–36 h after treatment, the average total residue level being 0.53 µg/mL [44]. A similar residue overview after FBZ administration to dairy cows in three different formulations has been reported [33]. The FBZ paste formulation produced the highest total residue level (0.32 µg/mL), followed by feed top dressing (0.26 µg/mL) and oral drench (0.16 µg/mL). Although OFZ was administered by the oral route [35], the milk residue concentrations showed a fate similar to that obtained after FBZ administration in dairy cattle. OFZ was the highest residue found in milk, followed by the FBZSO2 metabolite, and the parent drug was recovered at the lowest concentration in milk, which is consistent with the results obtained after FBZ treatment [33]. However, after oral administration of OFZ at the same dose as FBZ in dairy cows, the total analyte concentration in milk was the highest. Kappel et al. [44] also reported that a higher value of the total residue peak (0.53 µg/mL) was detected after oral administration of FBZ, probably due to a difference in the administered formulation.

In conclusion, after ABZ oral administration, the milk residue profile was different from that found after OFZ oral treatment at the same dose in dairy cows [35]. The faster metabolism of the ABZ parent drug relative to FBZ is consistent with these results. Moreover, the total milk residues found in milk after oral treatment of OFZ or ABZ were higher than those obtained after sc administration. Therefore, the pattern of milk excretion can also be related to the higher dose administered by the oral route.

Unlike other BZD compounds, the halogenated derivative triclabendazole (TCBZ) has excellent efficacy against juvenile and adult stages of *F. hepatica*. In veterinary medicine, TCBZ is extensively used for the treatment of fasciolosis. Both clinical and subclinical infections by *F. hepatica* are a major cause of lost production in dairy livestock, as evidenced by reductions in weight gain [45], fertility [46], and milk yield [47].

The control of *F. hepatica* is mainly based on anthelmintic treatments, but effective control of fasciolosis in lactating dairy cattle is difficult because of residues in milk; it can be treated only during dry-off periods, to avoid drug residues in milk. Although, fasciolosis is recognized as a serious problem in dairy animals in endemic areas, regular antihelmintic treatments are implemented. Therefore, the use of flukicides in dairy animals can result in undesirable residues in milk and dairy products, which can affect food safety [21,48].

The pharmacokinetics of TCBZ has been characterized in sheep [49], goats [50], and cattle [48,51]. The triclabendazole sulphoxide (TCBZSO) and triclabendazole sulphone (TCBZSO2) sulpho-metabolites recovered from the bloodstream of treated animals are not detectable in plasma after oral administration. This could be explained by the metabolic fate of TCBZ in the intestinal mucosa or liver [49,52]. Overall, enzymatic systems were found to be involved in the sulphoxidation and sulphonation of TCBZ. The sulpho-metabolites present in the bloodstream can diffuse in a wide range of tissues. Sulpho-metabolites are excreted through bile (45%) and urine (6.5% of the administered dose). Another excretion route is milk, reflecting what happens in the bloodstream. After oral treatment (12 mg/kg) of dairy cows, the proportion of TCBZSO and TCBZSO2 recovered in milk was between 0.11% (TCBZO) and 1.5% (TCBZSO2) of the administered dose. Therefore, TCBZSO2 was the most important residue recovered in cow’s milk [48,53,54]. Conversely, both sulpho-metabolites in milk occurred at lower concentrations in dairy goats [50] than in dairy cows, and none of them was detectable after 7 days because of the limit of quantification of the method (0.1 µg/mL).

At a value higher than 99%, TCBZ metabolites are strongly bound to plasma proteins. The extended persistence of TCBZ metabolites in the bloodstream (over120 h after oral treatment) is evidenced in sheep, dairy cows, and goats by longer plasma residence times and elimination half-lives (ranging from 19 to 49 h) compared with ABZ, FBZ, and their metabolites [55,56].

The concentrations of TCBZSO and TCBZSO2 in milk were similar to those found in plasma, and long persistence of TCBZSO2 (over 6 days) after oral treatment was evidenced in dairy goats and cows [48,50,54]. However, the maximum milk residual concentrations of sulpho-metabolites were smaller (approximately 50 and 17-fold, respectively) than plasma maximum concentrations. Moreover, the area under the concentration–time curve (AUC) ratios between plasma and milk (around 19-fold higher plasma than milk) showed a limited distribution of the sulpho-metabolites from the bloodstream to the milk [48] and a limited excretion of TCBZSO2 (1.3%) through milk [48,53] compared with other anthelmintic drugs such as endectocide compounds in dairy cattle [57]. Considering the results reported and the MRL (10 µg/kg) established in the European Union [58], the use of products containing TCBZ is not recommended for administration to dairy cows during the lactation period due to the long persistence of TCBZ residues in milk. Moreover, up to 5 days post-treatment, the concentrations of keto-TCBZ (marker residue) in milk were higher than the MRL established for dairy cows [54]. An interesting alternative is to treat dairy cows during the drying off period (60 days before calving) by oral administration of TCBZ (12 mg/kg) [54], where the residues of TCBZ in milk post-calving are below the currently established MRL. Therefore, TCBZ is a suitable flukicide to use during the dry period. However, in endemic areas where regular antihelmintic treatments are provided during the lactating period, a long withdrawal period must be strictly observed before consuming the milk from the treated animals.

### 2.2. Macrocyclic Lactones

The avermectins and milbemycins are closely related 16-membered macrocyclic lactones (MLs). These compounds are used against endo-and ectoparasites [12,59]. Despite the chemical differences, both families share high lipophilicity, extraordinary potency, and prolonged persistence, and their potent broad-spectrum activity is a distinctive trait among antiparasitic drugs. Ivermectin (IVM) and MXD are the most commonly used MLs worldwide to control endo- and ectoparasites in livestock, and they are currently marketed as topical (cattle), sc (cattle and sheep), and oral (sheep and goats) formulations.

In the late 1990s, an avermectin derivative named eprinomectin (EPM) was developed, and it was characterized by its low excretion in milk compared with other MLs. EPM was approved for topical administration to dairy and beef cattle [60].

The characterization of plasma kinetics of different MLs has been well documented in animal species. Comparative disposition in plasma [61,62] and target tissues [63,64] has been reported. Disposition kinetics of MLS differ according to the animal species [65], animal breed [66], type of formulation [67], and route of administration [68,69], among many other factors. Moreover, the plasma and milk kinetic behaviors of different MLs such as IVM and MXD have been reported in dairy cattle [57,70], sheep [68,71,72], and goats [69,73,74]. The results indicate that milk excretion is an important route of elimination for lipophilic drugs such as IVM, and particularly for MXD, which invalidates their use in dairy animals. However, topical formulations of MXD are currently approved for use in dairy cattle in some countries without a milk withdrawal time. Conversely, IVM is not allowed for use in dairy animals whose milk is yielded for human consumption; however, due to its long persistence in milk and milk-derived products, its extra-label use should be considered in terms of human safety.

Previous studies have shown that the routes of administration have a serious effect on partition milk–plasma behavior. After sc or oral treatment, IVM and MXD were detected in plasma and milk for a long period. After its sc administration (0.2 mg/kg), IVM was detected between 23 and 30 days in the milk obtained from dairy sheep, goats, and cattle [20,57,71,73]. However, after oral administration (0.2 mg/kg) to dairy sheep, IVM was detected in milk for a shorter period (up to 11 days post-treatment) [72].

MXD was detected in milk between 35 and 40 days after its sc or oral administration (0.2 mg/kg) to dairy sheep and goats [68,72,74]. The concentrations of IVM and MXD measured in milk (both treatments) in different species were greater than those recovered in plasma at all the sampling times. The greater endectocide plasma concentrations obtained after sc administration accounted for the higher values of IVM and MXD in milk.

However, some differences should be highlighted; the total IVM milk disposition in dairy sheep (127.6 ng.d/mL) after sc treatment was higher than that reported in goats (60.7 ng.d/mL). Similarly, the total MXD milk disposition after sc and oral treatments of dairy sheep was 1.71-fold higher than that reported in goats [74]. The permanence of higher concentrations of MXD in milk after sc administration accounted for the prolonged milk depletion half-life of the MXD (between 12 and 23 days) in different species. As reported, after sc treatment of dairy sheep, MXD persists 9.27-fold longer than IVM in milk. Moreover, the highest MXD tissue residue levels have been observed in sheep with higher fat content and greater body weight [63].

The extensive distribution of both endectocide drugs from the bloodstream to milk is clearly reflected in the partitioning milk/plasma ratio for IVM, which is 1 in goats and between 1.8 and 2.6-fold in sheep after sc and oral treatment, respectively. The percentage of total IVM dose excreted in milk after sc treatment reported in goats was estimated to be 0.3% [73], which is lower than in dairy sheep (between 0.7% and 0.8% in dairy sheep) [71,72] and dairy cows (5.46%) [57].

The partitioning of MXD into sheep’s milk is higher than that of IVM with milk/plasma concentration ratios between 10 and 18-fold [20,72]. The high lipid content of this milk, as well as the chemical structure of MXD, contributes to the extreme lipophilicity of this compound, making it more lipophilic than IVM [12],which may explain the higher partition of MXD than that of IVM to milk and longer persistence. As shown in Figure 1, the total percentage of the dose recovered in sheep´s milk for MXD after sc and oral treatments was significantly higher (*p* < 0.01) than that recovered for IVM. After sc treatment, MXD showed the highest percentage of the dose excreted in milk. However, the percentages of MXD and IVM excreted in milk after sc treatment were, respectively, 3.11 and 4.27 times higher compared to oral treatment. In the same way, prolonged detection of higher MXD tissue residue levels has been observed in sheep with higher fat content and greater body weight [63]. The fraction of the MXD dose eliminated in sheep´s milk (6.51% after sc and 2.09% after oral treatment) [72]was lower than that reported in goats (22.5% after sc and 5.71% after oral treatment) [74], but higher than that estimated in cows, where a suckling calf received about 5% of the dose excreted in milk after sc treatment [75].

The topical formulation of MXD is currently approved for use in dairy cattle in some countries with a zero-day milk withdrawal period. An MRL for residues of MXD (40 µg/kg) in the milk of bovine and ovine species has been established in the European Union [76]. Although licking behavior may drastically enhance milk residues of topically administered MXD, due to drug ingestion by licking, the MXD concentrations recovered at any time post-treatment did not exceed the permissible residual concentrations. As shown in Figure 2, the licking restriction period of about 5 days markedly influenced both plasma and milk concentrations of MXD. The availability of MXD in plasma and milk up to 5 days post-treatment was higher in the free-licking group compared with licking-restricted dairy cows. However, after 5 days, when the licking was permitted, the MXD total availabilities in plasma and milk up to 15 days were similar in both groups. Therefore, the percentage of MXD recovered in milk was similar.

The partitioning milk–plasma ratio for MXD after pour-on treatment (0.5 m/kg) to dairy cows was 1.3-fold, and the percentage of total MXD dose excreted in milk was estimated to be 0.2% [70]. The percentage of MXD excreted in milk after pour-on treatment reported is lower than that estimated in cows, where a suckling calf received about 5% of the dose excreted in milk after sc injection treatment [75]. However, the sway of licking on the pattern of milk residues should be considered, due to the high individual availability variation, particularly for those ML compounds whose approved MRL values in milk are much lower than those of the MXD MRL. Thus, the MRL for IVM (10 µg⁄kg) [77] in bovine milk is lower than that established for MXD, which could signify a greater risk if topically treated dairy cows ingest the drug by licking.

Compared to other MLs, EPM has a substantially reduced distribution in milk, as a consequence of minor changes introduced into the chemical structure of the avermectin molecule [60]. It was approved as a topical formulation for use in dairy ruminants with a zero-day milk withdrawal period. An MRL has been established in the European Union for residues of EPM (20 µg/kg) in the milk of all ruminant species [78].

The plasma and milk kinetic behaviors of EPM as a topical formulation have been investigated in dairy cattle [60,79,80], goats [81], and sheep [82,83]. The results about EPM excretion in different species indicate that the milk excretion is not an important route of elimination for EPM unlike other endectocide drugs such as IVM and, particularly, MXD. After pour-on treatment (0.5 mg/kg), a lower bioavailability was reported in dairy goats (8.24 ng·d/mL) and sheep (between 14 and 16 ng·d/mL) than in dairy cattle (between 91 and 239 ng·d/mL). These differences among species could be attributed to different factors such as the metabolic process, the amount of body fat, and skin layer morphology, among others [81,84]. Moreover, the EPM pour-on formulation is characterized by greater inter-individual variability in relation to low bioavailability and therefore a higher risk of underexposure.

As shown in Table 1, the concentrations of EPM reported in sheep, goat, and cattle milk are lower than those obtained in plasma. This low affinity of EPM for milk was reported in different species after topical administration (0.5 mg/kg); it was depicted by the low milk/plasma partitioning values in cattle (between 0.1 and 0.2-fold) [60,79], sheep (between 0.6 and 0.8-fold) [82,83], and goats (0.1-fold) [81]. These ratios were below 1.0 compared with the value obtained for other endectocide compounds. Therefore, a small fraction of EPM—between 0.03% and 0.3% of the total dose—was excreted in the milk of the different species reported.

Previous studies have shown that the routes of administration have a great effect on the pharmacokinetic behavior and efficacy of an endectocide [85,86]. It has also been reported that the peak IVM and MXD concentration in the milk of non-lactating cattle after oral treatment was higher than that obtained after topical treatment [66,87].As regards efficacy, a previous study on non-lactating goats receiving EPM at the recommended dose for cattle (0.5 mg/kg) showed that its efficacy against intestinal nematodes was lower [88,89,90]. Therefore, the selection of the optimum dose regimen and route of administration in different species is relevant to achieve good efficacy, prevent the development of resistances, and maintain safe levels of milk residues. The relationship between the plasma and milk kinetic behaviors of EPM and using EPM as a topical formulation at twice the recommended dose (1.0 mg/kg) has been investigated in dairy goats [81] and dairy sheep [83]. Moreover, the plasma and milk kinetic behaviors of EPM after treatment by non-approved routes (sc and oral treatment) in dairy goats [91,92] and cattle [80,93,94] have been investigated (see Table 1).

Some dates reported in dairy cattle (Chinese Holstein) have shown a similar bioavailability of EPM in plasma after pour-on (0.5 mg/kg) or oral (0.2 mg/kg) treatment. The maximum concentration values of EPM in milk after topical and oral treatment [80] were lower than those obtained after sc treatment (0.2 mg/kg) [94]. The milk-to-plasma ratio values obtained from Chinese Holstein after topical (0.124-fold) and oral (0.104-fold) treatment were similar to the value obtained from dairy cattle after topical (0.5 mg/kg) (0.102-fold) treatment [79].

In dairy goats, a significantly lower systemic bioavailability compared to cattle with a pour-on EPM formulation has been reported. Previous data on the efficacy and behavior kinetics of topical EPM under experimental conditions indicated that a higher dose rate had to be applied in goats twice the recommended dose (1.0 mg/kg) instead of 0.5 mg/kg [81,88,89,90,95]. Host physiology must also be taken into account because of a reduced availability of EPM in lactating compared to dry goats, probably related to a marked decrease in body fat reserves [81,96].

After pour-on treatment at the EPM dose of 1.0 mg/kg in dairy goats, a significantly higher systemic bioavailability has been reported compared to a dose of 0.5 mg/kg. The milk-to-plasma ratio values obtained from dairy goats were 0.12 and 0.25-fold after topical application of 0.5 or 1.0 mg/kg, respectively [81]. These values were lower than those obtained from dairy sheep at the same doses [83]. Moreover, a higher systemic bioavailability and mean residence time were reported in non-lactating goats [81].

Other routes of administration of EPM have been studied to ensure a suitable anthelmintic efficacy in goats. It has become known that sc and oral routes of administration lead to higher drug concentrations in plasma and tissues and higher efficacy compared with topical administration [91,97]. Badie et al. [92] have reported dates to characterize the EPM kinetics in plasma and milk after oral administration (0.5 or 1.0 mg/kg) of the topical formulation of EPM in lactating or non-lactating goats. A significantly higher systemic bioavailability has been reported for both doses after oral treatment compared with pour-on administration [81]. The elimination pattern of EPM in milk after oral administration was consistent with the plasma levels. The milk-to-plasma ratio values obtained from dairy goats were 0.36 and 0.33-fold after oral administration of EPM at 0.5 or 1.0 mg/kg, respectively [92]. These values were higher than those obtained from dairy goats at the same doses after pour-on treatment [81], and the percentage of total EPM dose excreted in milk was between 0.33% and 0.42% [92].

The results reported in dairy sheep, goats, and cattle have shown that following topical (0.5 or 1.0 mg/kg), oral (0.2 or 0.5 or 1.0 mg/kg), or sc (0.2 mg/kg) administration, milk excretion is not an important route of elimination for EPM, unlike other endectocide drugs such as IVM and, particularly, MXD. Therefore, the concentrations of EPM in milk were below the MRL established for the milk of all ruminant species (20 µg/kg) [78], although a non-approved route of administration was used.

**Table 1 animals-11-02758-t001:** Mean (±SD) peak concentrations and ratios of the area under the concentration vs. time curves in milk and plasma for eprinomectin administered either orally or topically in lactating dairy ruminant species.

Species	Route of Administration	Dose (mg/kg)	C_max_ Plasma (ng/mL)	C_max_ Milk (ng/mL)	Ratio AUC Milk/Plasma	Reference
cattle	pour-on	0.5	43.76 ±18.23	5.14 ± 2.53	0.102 ± 0.048	[79]
		0.5	16.16 ± 6.02	2.28 ± 0.85	0.124 ± 0.041	[80]
	oral	0.2	30.02 ± 5.73	3.14 ± 0.88	0.104 ± 0.022	[80]
	subcutaneous	0.2	44.0 ± 24.2	6.4 ± 1.8	0.16 ± 0.01	[94]
sheep	pour-on	0.5	2.28 ± 0.41	1.5 ± 0.32	0.69 ± 0.08	[82]
		0.5	2.22 ± 0.88	1.37 ± 0.55	0.79 ± 0.12	[83]
		1.0	5.25 ± 2.71	7.07 ± 5.16	1.12 ± 0.43	[83]
goat	pour-on	0.5	2.20 ± 0.52	0.32 ± 0.08	0.122 ± 0.07	[81]
		1.0	2.98 ± 1.37	0.82 ± 0.08	0.254 ± 0.179	[81]
	oral	0.5	15.48 ± 6.64	5.34 ± 2.23	0.36 ± 0.05	[92]
		1.0	38.10 ± 8.57	11.47 ± 2.23	0.33 ± 0.08	[92]

C_max_: peak plasma or milk concentration; AUC: area under the concentration vs. time curve where the drug was measurable.

### 2.3. Salicylanilides

Closantel, oxyclozanide, and rafoxanide are the most extensively used salicylanilides for the treatment of liver fluke [98,99]. Most salicylanilides are strongly bound to plasma proteins (more than 99%), which prolongs the elimination half-lives and the presence of drug levels in plasma and tissues [100].

Closantel (CLS) is highly useful in the treatment of adult flukes as oxyclozanide (OXZ), and it shows good activity against immature flukes aged 6–8 weeks but is not effective against earlier stages [101]. Moreover, CLS is effective against bloodsucking nematodes, and the larval stages of some arthropods that feed on blood and plasma. CLS is currently used in strategic control programs of haemonchosis as an alternative drug for the treatment of BZD and levamisole-resistant strain of *Haemonchus contortus* [22]. Moreover, it is useful for the treatment of adult nematodes in both sheep and cattle, as well as against certain ectoparasites and the nasal bot *Oestrus ovis* in sheep. Rafoxanide (RFX) has been extensively used worldwide against fasciolosis in sheep and cattle. It is active against gastrointestinal nematodes and nasal bot fly. At a therapeutic dose, RFX is highly effective in the treatment of adult flukes and shows good activity against immature flukes aged 4weeks at elevated doses [101].

At present, a small number of salicylanilide compounds are registered to be used in animals whose milk is meant for human consumption. For these compounds, after treatment, a withdrawal time is necessary to avoid residual concentrations exceeding the permitted maximum residue limit (MRL). In the European Union, an MRL has been established for residues of CLS (45µg/kg) in the milk of bovine and ovine species [102] and of OXZ (10 µg/kg) in the milk of all ruminant species [103]. Use of RFX for treating lactating dairy animals is not allowed, and the provisional MRL (10 µg/kg) in milk has expired [104]. 

CLS is a weakly acidic molecule and is particularly lipophilic. It is formulated for oral, intraruminal, or parenteral administration in ruminants. Moreover, CLS is an extra-label used in different dairy production systems. CLS is widely bound to plasma proteins (>99%) and has a long terminal half-life of 14.3–14.5 days in sheep [100,105] and 8.9 days in dairy goats [106]. As a consequence of its high plasma protein binding, the duration of therapeutic levels of CLS in plasma is prolonged, which limits the distribution to tissues [107,108]. The elimination half-life being shorter in dairy goats [106] compared with other animal species [100,105] could be attributed to the differences in the turnover of albumin.

CLS is poorly metabolized in the liver, with 90% of the compounds corresponding to the parent drug [107], and is excreted through feces (80%) and urine (<0.5%). Some works with other antiparasitic drugs such as OFZ [109,110] and ABZ [29] have demonstrated faster hepatic metabolism and clearance in goats. Similarly, CLS in goats showed a shorter elimination half-life (9 days) [109] compared to that reported in sheep (14 days) by Hennessy et al. [105].

Another minor route of excretion of CLS is milk. In dairy cattle, approximately 1% of the administered dose was found to be excreted through milk per day, and the partitioning ratio of plasma-to-milk concentration was in the order of 50:1 [102]. Similarly, after oral administration of CLS (10 mg/kg) in dairy goats [106], the milk concentrations reported were significantly lower and persistent compared to those found in plasma. The results reported show that even though the partition of CLS from the blood to the milk is a limited amount, most of the drug persists in the bloodstream. The mean plasma C_max_ and AUC values were approximately 44 and 66-fold higher, respectively, compared to those observed in milk. This is clearly depicted by the low volume of distribution during the elimination phase and the AUC milk/plasma ratios (0.015), and a low percentage of the dose was recovered in the milk (1.65% of the administered dose). As shown in Figure 3, the concentration of CLS in milk up to 29 days after oral administration (10 mg/kg) in dairy goats was higher than the MRL established (45µg/kg) for the milk of bovine and ovine species [102]. Therefore, the withdrawal time for CLS in goat’s milk is long; it was calculated to be between 39 and 43 days [106]. Similarly, Power et al. [111] reported that the milk residues of CLS were greater than those established for MRL 52 days after its subcutaneous administration to dairy cattle.

Another member of the salicylanilides is OXZ; it is an exclusive flukicide licensed for use in lactating dairy ruminants, and it is marked as an oral drench containing only OXZ or combined with levamisole hydrochloride or OFZ. Following oral administration to sheep (15 mg/kg), OXZ is extensively (greater than 99%) bound to plasma protein. Its terminal half-life in sheep (6.4 days) is shorter than that of other salicylanilide compounds such as CLS (14.5 days) [100]. However, after oral administration of OXZ (15 mg/kg) in combination with levamisole (7.5 mg/kg) as a tablet, the terminal half-life of OXZ in sheep (21.7 h) was shorter than that of OXZ administered alone [112]. In dairy cows, after oral administration of OXZ (10 mg/kg) in combination with levamisole, OXZ was rapidly excreted through milk. The highest residual levels in milk were reported at the third milking, and OXZ concentrations in milk were below 10 µg/kg (MRL value approved) at the 8th milking. However, the inter-individual variation was high (between zero and 8th milking) [113]. Another author has reported that after oral administration (10 mg/kg), OXZ residues in cow´s milk were detected until 30–47 h above 1 µg/kg, and the OXZ residues in milk were below the MRL at all sampling times and OXZ residues [114].

RFX products are currently marketed in the European Union for the treatment of cattle and sheep. Following oral dosing (12 mg/kg), RFX is well absorbed, reaching the maximum concentration in plasma between 24 and 48 h, but it is slowly excreted. The elimination half-life ranges from 5 to 10 days due mainly to its high protein binding [115]. Compounds against other parasites, such as nematodes and cestodes, are included in the formulation of some marketed fasciolicides. IVM (1%) plus RFX (12.5%) as an injectable solution has been licensed for the treatment of endo-ectoparasites in cattle, sheep, and goats [116]. After sc administration of this combined product in calves and sheep, enhanced IVM AUCs and prolonged elimination half-lives were observed. High residues of RFX in milk were observed after oral administration (11.75 mg/kg) to lactating cows, with the maximum (516 μg/kg) showing 3–4 days after treatment. However, when RFX was orally administrated to dairy cows at the start of the dry period, the means of concentrations were lower than 10 µg/kg after the colostrum period (3 days after calving)[104].

### 2.4. Miscellaneous: Nitroxynil and Clorsulon

Nitroxynil (NTX) is a nitrophenolic compound. It is highly effective against the adult stage of *F. hepatica* and is not effective against young flukes (less than 6 weeks). Moreover, it is used for the control of IVM-resistant and BZD-resistant *H. contortus* in sheep. NTX is more effective when administered by parenteral routes. Formulations containing NTX plus IVM or clorsulon or CLS are marked for administration in ruminants. NTX is rapidly absorbed after its sc administration and is bound strongly to plasma proteins (98%) [117]. NTX is extensively metabolized, and in all species, the residues in plasma were higher than the residues in tissues and consisted almost entirely of NTX [118]. The primary route of elimination is urine, although it is also excreted through feces and milk. In the European Union, an MRL has been established for residues of NTX (20 µg/kg) in the milk of bovine and ovine species [118]. The NTX residues were found in milk after sc treatment (10 mg/kg) of dairy cows at the beginning of the dry period (60–80 days before calving). However, after the colostrum period (3 days after calving), the concentration of NTX residues in milk was lower than the MRL value established (10 µg/kg) [118,119]. Moreover, Whelan et al. [119] have shown that after sc administration of NTX to lactating dairy cows, the milk residues persisted for 58 days. In general, when the dry period is longer (60–80 days) between treatment and calving periods, the level of NTX residues in milk is below the MRL value established.

Clorsulon is a benzene-disulphonamide derivative. It is effective against adult flukes in sheep and cattle when given as an sc injection at a dose rate of 2 mg/kg, and at a dose rate of 4–8 mg/kg it is highly effective against flukes aged 8 weeks. After oral treatment at a dose of 7 mg/kg, it is highly effective against adult flukes. Clorsulon is available as oral drench (sheep and cattle) and injectable (cattle) formulation with IVM alone or NTX plus IVM for simultaneous treatment (*F. hepatica* and nematode infection). Moreover, it was shown to be highly effective against immature flukes (2 and 4weeks) [120].

The major residue recovered in milk was the unchanged drug; about 0.7% of the dose was recovered in milk during a period of 7days. After sc administration of clorsulon plus IVM (3 mg/kg plus 0.3 mg/kg) to lactating dairy cows, milk residues depleted rapidly and the depletion half-life in milk was found to be 31 h [121]. However, use of clorsulon for treating lactating dairy animals is not allowed, and its provisional MRL (16 µg/kg) in bovine milk has expired [121]. The use of clorsulon is not permitted in animals that produce milk for human consumption.

## 3. Milk Excretion: Residues in Milk-Derived Products

The risks associated with chemical residues in milk, meat, eggs, etc., may be present after treating food animals with different drugs, including antiparasitic compounds. In relation to this, the Food and Drug Administration (FDA) has developed a multicriteria model for risk evaluation of animal drug residues in milk and milk products. This is an important step to re-evaluate which animal drug residues should be considered in the milk-testing programs. It is widely known that the beta-lactam antibiotics are commonly used in dairy cows; however, other kinds of drugs are also administered to dairy animals with antiparasitic drugs, especially avermectins, the highest ranked drug classes [122].

Therefore, the fate of residues is particularly important when raw milk is subjected to different industrial processes (heating, cooling, clotting, cooking, etc.). Many scientific reports have shown that the residues of antiparasitic drugs such as IVM, MXD, and EPM in raw sheep´s milk are stable during conventional milk heating processes (65 °C, 15 s; 75 °C, 15 s; 90 °C, 30 min) and the fermentation process extensively used in the dairy industry [19,123,124]. As regards ABZ, sulpho-metabolites and amino sulphone residues in cattle´s milk have been shown to be stable at conventional milk heating processes [38]. Moreover, TCBZ, a halogenated BZD, has been shown to be stable at a high temperature (185°C) during milk powder manufacture [21]. CLS residual concentrations were stable during thermal processing of goat milk [125]. Similarly, Power et al. [111] also showed that CLS residues in cattle milk were heat stable to pasteurization and spray-drying temperatures (185 °C).

Considering that the antiparasitic drugs reported were stable during conventional heating treatment, the levels of residues detected in raw milk could be directly applicable to estimating consumer exposure and dietary intake calculations when consuming heat-processed fluid milk. However, after milk processing to obtain milk products such as cheese, yogurt, ricotta, and butter, the residues of antiparasitic drugs were higher than those measured in the milk used for production. Many scientific reports have shown that endectocide drug residues in milk products such as cheese are higher than those in milk.

A high proportion of parent IVM and MXD (between 2.4 and 2.8-fold) was found in curd while processing sheep’s milk due to the higher fat content of this milk product (curd) and drug lipophilicity [12,20]. Similarly, in mozzarella cheese, the residue of IVM was 4-fold higher than that obtained in milk [18]. Moreover, the concentrations of IVM and MXD gradually increased during the ripening period, and the highest residual concentrations for IVM and MXD were obtained after cheese maturation (40 days) [19,20]. The mean cheese/milk ration for MXD was 3.4-fold and for IVM it was between 3.3 and 4-fold; these results were correlated with water loss and the enhancement of fat and solid contents in the ripened cheese [19,20]. Therefore, a lower proportion of IVM (between 0.1 and 0.16-fold) and MXD (0.16-fold) ended in the whey (whey/milk ratio) due to the high water content in whey [19,20]. Although IVM and MXD residues in sheep´s milk were higher than those obtained for EPM, the mean ratio between curd and milk for EPM (3.4-fold) was similar to that obtained for both IVM and MXD [19,20,82]. As shown in Figure 4, the highest residual concentrations of EPM were obtained in ripened cheese (40 days) made with the milk obtained from treated sheep [82] and goats [126]. The EPM residues in cheeses were higher in goat cheeses, possibly as a consequence of greater loss of water. However, these concentrations in cheese were below the MRL (20 μg/kg) permitted for all ruminants’ milk.

Conversely, antiparasitic drugs that are less lipophilic than the sulpho-metabolites of ABZ (ABZSO and ABZSO2) showed a high concentration in whey (70%), and a lower concentration in cheese was elaborated with cow´s milk after treatment. However, the residual concentrations in cheese were higher than those measured in the milk used for its production. ABZSO2 metabolites showed the highest residual concentration after ripening cheese [37,38].

The sum of TCBZ and its metabolites, expressed as keto-TCBZ residues, was higher in cheese (5-fold), butter (9-fold), and skim milk powder (15-fold) than that in milk from cows after treatment. Moreover, TCBZ residues were detected in milk products, even though there was no detectable residue in the milk used to manufacture these products [21]. Moreover, the mean cheese/milk ratio of TCBZ sulpho-metabolite concentrations was 13.4-fold higher in cheese elaborated with milk obtained from treated dairy cows [48].

CLS residual concentrations in cheese (3-fold) and ricotta (6-fold) were higher than those measured in goat´s milk used for their production. Moreover, the CLS concentration in ricotta was 20-fold higher than that in the whey used for its production [106]. Similarly, Power et al. [111] reported high presence of CLS residues in milk products (cheese, cream, and butter)—between 6 and 10-fold higher compared with the milk obtained on days 2 and 23 after its subcutaneous administration to dairy cattle.

A different behavior was observed for other members of the salicylanilide group. OXZ residues in soft (3 days ripening) and hard cheeses (35 days ripening) were lower than those detected in the cow’s milk used for their elaboration. The OXZ residues in both types of cheese were lower than those in milk, unlike the whey, in which the highest concentrations of OXZ were found—10-fold higher than those in milk due to the strong binding of OXZ to whey’s proteins [113]. Therefore, in view of the reported data, the most fat-soluble antiparasitic drugs, such as the endectocides IVM, MXD, and EPM, and flukicidal CLS and TCBZ, among others, have shown the highest residual concentrations in high-fat milk products, such as cheese, cream, and butter, due to their lipophilicity and the enhancement of fat and solid contents in these milk products.

## 4. Conclusions

Although different management strategies are implemented to minimize the production losses due to parasitic diseases, the use of chemicals (antiparasitic drugs) is still the main tool available for dairy and beef animals. Therefore, after treating with antiparasitic drugs, risks associated with residues in milk may be present. There is solid scientific evidence showing that most of the residues from antiparasitic drugs are chemically stable during conventional milk heating and fermentation processes that are extensively used in the dairy industry. Thus, the fate of antiparasitic residues is very important when raw milk is subjected to different industrial processes. Furthermore, experimental data are available that quantitatively describe changes in the partitioning/distribution of the drugs during milk processing. The residual concentrations of lipophilic drugs in milk products tend to be higher compared to those measured in raw milk collected from treated animals.

The pharmacokinetics-based information on the plasma-to-milk drug exchange and pattern of milk excretion described in this review article clearly indicates that the highest adverse/risky impact for the consumer may occur in antiparasitic molecules with some of the following features: (i) high partition coefficient of plasma to milk (highly lipophilic), (ii) long persistence of milk residual concentrations, and (iii) chemically stability to heat degradation or water removal after milk processing. These properties identify the drug molecules reaching the highest residual concentration in high-fat milk products, such as in the case of the broad-spectrum ML antiparasitics. Rational anti-infective pharmaco-therapy in dairy animals should include careful attention to all the factors expected to result in increased exposure of consumers to drug residues in milk and derived products. The scientific information reviewed here provides support to avoid the undesirable consequences of the presence of residual drug concentrations in milk and derived products intended for human consumption.

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
