# Peer review of "The Pattern of Blood–Milk Exchange for Antiparasitic Drugs in Dairy Ruminants"

_animals, 2021, doi:10.3390/ani11102758_

Round 1
Reviewer 1 Report
- Figure 1 and 2, Have the authors compare their data with others data present in literature? if no I suggest to show in figure 1 and 2 the summary of finding from different studies. Otherwise the authors should clearly stated that this review is based only on their previously studies.
- for each drug class the authors should discuss the main findings for each animal species
- figure 4, why the authors show only data from sheep milk ?
Author Response
"Please see the attachment"

Reviewer 2 Report
This article entitled “The pattern of blood-milk exchange for antiparasitic drugs in dairy ruminants” have certain merits, due to the importance of the traceability of antiparasitics from their administration in animals to the consumer.
However, there are a number of minor changes, which should be incorporated.
In general, you should review the units (mg / mL) referring to drug residues in milk (µg / mL that appears in some of the consulted publications).
The European regulation regarding MRLs should also be incorporated in the bibliography.
.- REGULATION (EC) No 470/2009 OF THE EUROPEAN PARLIAMENT AND OF THE COUNCIL of 6 May 2009 and
.- COMMISSION REGULATION (EU) No 37/2010 of 22 December 2009 on pharmacologically active substances and their classification regarding maximum residue limits in foodstuffs of animal origin (Text with EEA relevance)
Others comments
Line 39. The absorption, distribution, metabolism and elimination are the physiological processes
The authors should change the term elimination to excretion because elimination includes the processes of metabolism and excretion, and pharmacokinetic processes better than physiological processes??
Line 118, avoid residual concentration above of the MRL estipulate or zero concentration for drugs without MRL established.
Add reference mentioned above
Line 231 … and none of them was detectable after 7 days.
The level of detection depend of the limits of quantification (LLOQ) which should be indicated
Line 501 CLS showed a shorter elimination half-life of CLS [108] compared to those reported in sheep by Hennessy et al. [104)
Authors should indicate the half-life value(s).
Figure 3 CLS showed a shorter elimination half-life of CLS [108] compared to those reported in sheep by Hennessy et al. [104
To facilitate the understanding of Figure 3, the authors should incorporate the MRL value of 45µg / kg in the figure captions.
Author Response
"Please see the attachment"
